

# Marine Heatwaves in the Mediterranean Sea: A Convolutional Neural Network study for extreme event prediction

Antonios Parasyris[1], Vassiliki Metheniti[1], Nikolaos Kampanis[1] and Sofia Darmaraki[1]

[1]*Foundation for Research and Technology-Hellas, Institute of Applied and Computational Mathematics, 70013 Heraklion, Greece*

**\*Correspondence to: Antonios Parasyris ([antonisparasyris@gmail.com](mailto:antonisparasyris@gmail.com))**

**Abstract**: In recent decades, the Mediterranean Sea has experienced a notable rise in the occurrence and intensity of extreme warm temperature events, referred to as Marine Heatwaves (MHWs). Hence, the ability to forecast Mediterranean MHWs in the short term is an area of ongoing research. Here, we introduce a novel machine learning (ML) approach, specifically tailored for short-term predictions of MHWs in the basin, using an Attention U-Net Convolutional Neural Network. Trained on daily Sea Surface Temperature anomalies and gridded fields of MHW presence and absence between 1982-2017, our model generates a spatiotemporal forecast of MHW occurrence up to 7 days in advance. To ensure robust performance, we explore various configurations, including different forecast horizons and U-net architectures, number of input days, features, and different subset splits of train-test datasets. Comparative analysis against a Persistence benchmark reveals that our model outperforms the benchmark across both forecast horizons. For the 7-day forecast, the model achieves a 15% improvement in forecasting accuracy of MHW presence over the Persistence, while for the 3-day forecast, this improvement percentage drops to 4.5%. Notably, the discrepancy between our model and the benchmark narrows for shorter horizons, as the Persistence method also achieves high accuracy in the 3-day forecast. Our proposed ML methodology offers a data-driven alternative for MHWs prediction with reduced computational requirements, which can be applied across different regions of the global ocean, providing relevant stakeholders and management authorities with essential lead time for implementing effective mitigation strategies.

**Keywords:** Marine Heatwaves; Mediterranean Sea; Machine learning; Convolutional Neural Networks; Extreme event forecasting; U-net, Attention

## 1 Introduction

Since the 1980s, the Mediterranean Sea has undergone a mean SST increase twice that of the global average, ranging from 0.035 to 0.041 °C/year (Pisano et al., 2022). As a consequence, Mediterranean MHWs have increased in frequency, intensity and spatial coverage, with profound disruptions on marine ecosystems and communities that rely on them (Smith et al., 2021); Specifically, MHWs have caused numerous mass mortalities of native and the migration of invasive species in the Mediterranean Sea (Garrabou et al., 2022), threatening the region's rich marine biodiversity and commercially valuable fish stocks (Lacoue-Labarthe et al., 2016). Consequently, the ability to forecast MHWs has become central to the field of extreme oceanic events in the



basin, as it enables the development of proactive measures for the mitigation of subsequent and potentially adverse effects on marine ecosystems and socio-economic activities.

Essential to the early prediction of MHWs are advanced monitoring systems and forecasting models, which have proven reliable (Balaji, 2021) on global (Konsta et al., 2023) and regional applications (Schultz et al., 2021), in addition to sophisticated climate models that offer a comprehensive understanding of MHW drivers by simulating the complex interactions between atmospheric conditions and local oceanic circulation patterns (Darmaraki et al., 2019). In the Mediterranean Sea, MHWs have been predicted in the short term by the regional Copernicus Mediterranean Forecasting System, which successfully captured phases of the summer 2022 event (Mcadam et al., 2023), while climate models have projected MHW frequency and characteristics throughout the 21st century, under different climate change scenarios (e.g. (Darmaraki et al., 2019; Konsta et al., 2023)). However, forecasting systems and state-of-the-art numerical models present significant challenges due to the inherent uncertainties and chaotic nature of the climate system in addition to the associated high computational costs.

Thus, there is a growing interest in the use of ML techniques, particularly data-driven, Deep Learning (DL) models, to produce short- and long-term forecasts more efficiently (Schultz et al., 2021; Balaji, 2021). DL models are used as surrogate models that overcome the computational constraints present in classical numerical weather prediction (NWP) models. By training on large observational and/or model datasets, DL models can generate an extensive ensemble of forecasts, enhancing and complementing traditional NWP models (e.g. (Chattopadhyay et al., 2020)). Compared to traditional methods, ML models typically encounter fewer issues with bias (Jacox et al., 2022) and are especially skilled at capturing and representing intricate and nonlinear dynamics in data (Hornik, 1991).

Current research predominantly focuses on the forecast of SST (Taylor and Feng, 2022), which allows the establishment of contextualized thresholds based on specific requirements and conditions. This approach requires additional processing and expertise to identify MHWs (Hobday et al., 2016), which may complicate the application of these predictions by end users and management operators, a challenge that can be circumvented through data-driven spatiotemporal MHW prediction algorithms e.g., (Sun et al., 2023). Globally, the forecasting of SST using shallow ML techniques, such as linear regression and various statistical methods, is not new and dates back to the 1970s (Anding and Kauth, 1970; Fdez-Riverola et al., 2002; Mcmillin, 1975). During the last decade, the field has seen a significant shift towards the application of DL methods. These include Recurrent Neural Networks, long short-term memory networks (LSTMs; (Xiao et al., 2019; Liu et al., 2018)), Convolutional Neural Networks (CNNs; (Han et al., 2019)), and hybrid methodologies that combine these techniques (Taylor and Feng, 2022). These advancements have enhanced the accuracy and capabilities of SST predictions.

In contrast, direct forecasting of MHWs has received less attention. For instance, Giamalaki et al. (2022), employed a Random Forests (RF) method and successfully forecasted spatiotemporal occurrence of MHWs in the North Pacific Ocean. However, their approach exhibited limitations in accurately predicting the intensity and duration of these events, underscoring the challenges in developing robust predictive models for such complex



phenomena (Schultz et al., 2021). More recently, Sun et al. (2023) advanced the field by training a hybrid model, that integrated CNNs with LSTM layers, to predict the occurrence and spatial development of MHWs. Their study proposed an innovative approach, combining CNN-LSTM architectures with a U-net CNN regression model to forecast both SST anomalies and binary classification maps, indicative of the presence or absence of MHWs in the future. These outputs served as indicators of MHWs when they exceeded certain thresholds. This hybrid method focused on predicting the spatial distribution of MHWs rather than generating simple time-series forecasts and demonstrated significant potential for predicting MHWs with a lead time of up to 7 days. Such predictive capabilities can be particularly useful for the implementation of timely and effective mitigation strategies in regions highly vulnerable to MHW impacts, such as the Mediterranean Sea.

Nevertheless, the use of ML techniques for the early prediction of Mediterranean MHWs is an emerging field of research. One of the pioneering efforts has used Artificial Neural Networks to predict the seasonal and inter-annual variability of SST in the western Mediterranean as well as the impact of the severe summer MHW of 2003 (Garcia-Gorriz and Garcia-Sanchez, 2007). By training on a variety of meteorological variables from 1960 to 2005, including 2-meter air temperature, wind, and sea level pressure, the study achieved reliable monthly SST predictions. More recently, Bonino et al. (2023b) evaluated the effectiveness of various ML algorithms, including LSTM networks, CNNs, and RF on the forecast of daily SST, with a weekly lead time across 16 Mediterranean sub-basins. Their study focused on predicting SSTs that exceeded specific thresholds, indicating the potential occurrence of MHWs. This approach successfully reduced computational costs associated with the processing of 2D temperature fields, albeit at the expense of detailed information on the spatial variability of anomalous SST within each sub-basin. The results of this study outperformed outputs from the Copernicus Mediterranean Sea Analysis and Forecasting System, a sophisticated model that provides daily forecasts of ocean variables (Coppini et al., 2023). A hybrid statistical downscaling method combined with Regional Climate System Model (RCSM) outputs has been also proposed by Doury et al. (2023), where a U-net CNN architecture reproduced accurately the complex, RCSM-based, near surface temperature patterns of the northwest Europe at a high resolution (12km). The ability of the proposed neural network to learn the relationship between the large-scale predictors and the local-scale variables at a low computational cost, holds considerable promise for exploring the uncertainties associated with the identification of ocean temperature extremes in future projections. Overall, the application of ML techniques has showcased considerable potential in advancing our comprehension and forecasting of MHWs in the Mediterranean Sea.

Here, we combine well-established methodologies and tools to create a novel configuration of an Attention U-Net CNN, specifically tailored for the forecast of MHWs in the Mediterranean Sea. Using a minimal set of RCSM variables, primarily the daily, spatial distribution of SST (anomalies) and spatiotemporal information on the presence and absence of MHWs in the basin, produced by a MHW identification algorithm, we train the Attention U-Net CNN model to forecast MHW occurrence 3 and 7 days in advance. This work is structured as follows: Sect. 2 describes the ML technique, the input variables that were used to train the U-Net model and the error metrics used for validation. Results on MHW forecasting, based on various neural network configurations are shown in Sect. 3 followed by the Discussion and Conclusions on Sect. 4.



## 2 Methodology

### 2.1 Study area

The Mediterranean is a semi-enclosed, transitional area surrounded by the temperate zone to the north and the subtropical zone to the south and east. Its complex topography features sharp mountains, mild coastlines and
desert regions, creating a region highly sensitive to climate change. The Mediterranean Sea is connected to the Atlantic Ocean through the Strait of Gibraltar to the west and to the Black Sea via the Bosphorus Strait to the northeast, which serves as the main freshwater inflow for the eastern basin. It has a mean depth of 1500m (Bethoux et al., 1999) and is divided in several sub-basins, including the Ionian, Tyrrhenian, Aegean, Adriatic, Alboran, Balearic, and Ligurian Seas (Fig. 1). The region is also characterized by a distinctive thermohaline
circulation driven by surface heat and water losses. This unique circulation system balances the excess evaporation over the Mediterranean, contributing to a net buoyancy flux towards the atmosphere, which plays a crucial role in the regional climate dynamics.

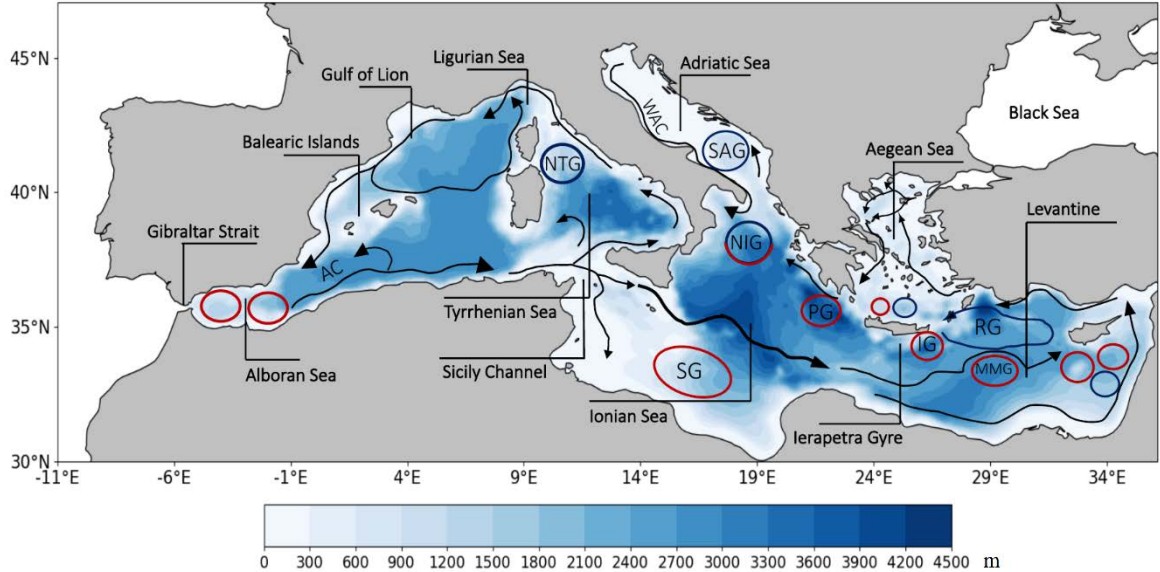

**Figure 1:** *The bathymetry, main circulation and sub-basins of the Mediterranean Sea. Key features of the surface circulation are shown with black arrows, with anticyclonic and cyclonic systems represented by red and blue circles, respectively. Acronyms: AC, Algerian Current; NTG, North Tyrrhenian Gyre; SG, Sidra Gyre; WAC, Western Adriatic Current; SAG, Southern Adriatic Gyre; NIG, Northern Ionian Gyre; PG, Pelops Gyre; IG, Ierapetra Gyre; RG, Rhodos Gyre; MMG, Mersa-Matruh Gyre. Bathymetry (m) is given in colors based on the CNRM-RCSM6 model. The figure is reprocessed based on (Menna et al., 2022; Velaoras et al., 2024; Darmaraki et al., 2024)*

### 2.2 Input Data

To identify surface MHWs in the Mediterranean Sea and train the ML algorithm for their short-term prediction, we obtain daily gridded, SST outputs from the fully-coupled, regional climate system model CNRM-RCSM6, that run on hindcast mode between 1982-2017 (Ruti et al., 2016). The model covers the entire Mediterranean Sea domain, has a 6-8 km horizontal resolution, with a varying vertical resolution, over 75 vertical levels in the ocean



(NEMOMED12, (Beuvier et al., 2012; Waldman et al., 2017)) and its lateral boundary conditions come from
ERA-Interim (Berrisford et al., 2009).

The ML method is trained, tested and validated on a combination of 2 types of input variables: (1) daily, gridded SST anomalies (SSTA) relative to the 1982-2017 period and (2) gridded fields of MHW presence/absence. The daily spatial coverage of surface MHW is computed using the updated version of the MHW detection algorithm
by (Hobday et al., 2016), available at https://github.com/coecms/xmhw. According to this definition, a MHW occurs when the local SST is above a 30-year climatological threshold of the 90[th] percentile of SST, for at least 5 consecutive days. The MHW identification method yields gridded, binary classification masks of daily MHW presence/absence (1/0) for the entire Mediterranean Sea domain, between 1982-2017. The predominant classification of grid points as MHW-absent (0) for most days of the year results in an imbalanced input dataset
that affects the forecasting accuracy of any data-driven model (Bonino et al., 2023b; Sun et al., 2023).

To achieve a more balanced input dataset and reduce memory requirements during training the gridded MHW occurrence fields are first downsampled. This process considers a non-overlapping 2x2 submatrix around a point and assigns label (1) in the presence of at least one MHW-affected pixel, or a label of (0) in its absence, leading
to a decreased spatial resolution of MHW occurrence fields. The same approach is followed for points in the immediate proximity to the coast. The final spatial resolution of the Mediterranean Sea resolves to 128x272 points. Despite the reduction in the spatial resolution, this approach increases the proportion of MHW presence to 14% (from the initial 7.7%) across the entire domain, while still preserving a significant portion of the complicated Mediterranean Sea features and the geographical location of each MHW. To match the spatial
resolution of the gridded MHW occurrence fields, we also downsample the daily SSTA of the Mediterranean Sea, by means of spatially averaging within a non-overlapping 2x2 submatrix around a point. Daily SSTA are further normalized to a range [0,1], aligning with the scale of the gridded MHW occurrence data, with a view to improve the prediction accuracy of the U-Net CNN (Xiao et al., 2019).

As an input to the CNN model for a given day, we insert the gridded fields of MHW occurrence and SSTA from preceding days, targeting the prediction of MHW spatial coverage 3 and 7 days ahead (Fig. 2). The limited set of input variables is selected as an effective approach to balance the risk of overfitting with computational efficiency in our method. This decision was further informed by a lagged correlation analysis, which revealed moderate correlations between SSTA and atmospheric variables such as air temperature, latent heat flux and shortwave
radiation, with SSTA displaying the highest lagged autocorrelation. We then perform sensitivity tests to assess the impact of varying the number of preceding days as input, specifically 0, 2 and 4 days for the MHW occurrence fields and 5, 10 and 14 days for SSTA, on the model's predictive ability. A cosine and sine functions are also inserted as input features, indicating the yearly seasonal cycle, assuming a unique combination of values for each day of the year, spanning the period from 1/1/1982 to 31/12/2017:



$$\cos_t = \cos\left(\frac{2\pi t}{365}\right); \forall t \in \{1, 2, \ldots, Ndays\}$$

$$\sin_t = \sin\left(\frac{2\pi t}{365}\right); \forall t \in \{1, 2, \ldots, Ndays\} \;,$$   (Eq. 1)

where the total number of days is denoted with *Ndays*.

## 2.3 The Attention U-net CNN model

In this study, we employ a data-driven approach, which means that the nature, quantity and quality of our data (e.g. number of available years, temporal discontinuities in data, spatiotemporal resolution and forecasted
variable characteristics including class imbalance, seasonality, randomness etc.) dictate the methodological choices for forecasting MHW occurrence across different time horizons. Leveraging the capabilities of a neural network architecture, enhanced with attention mechanisms (Attention U-Net CNN mode) which emphasize key features while suppressing noise, and capturing important spatiotemporal patterns, we focus on 3-day and 7-day forecast horizons. This is deemed a sufficient time for proactive decision-making by authorities and local
stakeholders before a MHW incident (Giamalaki et al., 2022).

### 2.3.1 Attention U-net Architecture

Here, the prediction of MHWs is considered a supervised classification/regression challenge, for which we employ the specific neural network architecture illustrated in Fig. 2, based on the U-net CNN architecture
proposed by Ronneberger et al. (2015). Due to the use of several intermediate (hidden) layers, the method is classified as a deep neural network category, with the architecture including both a contracting and expanding

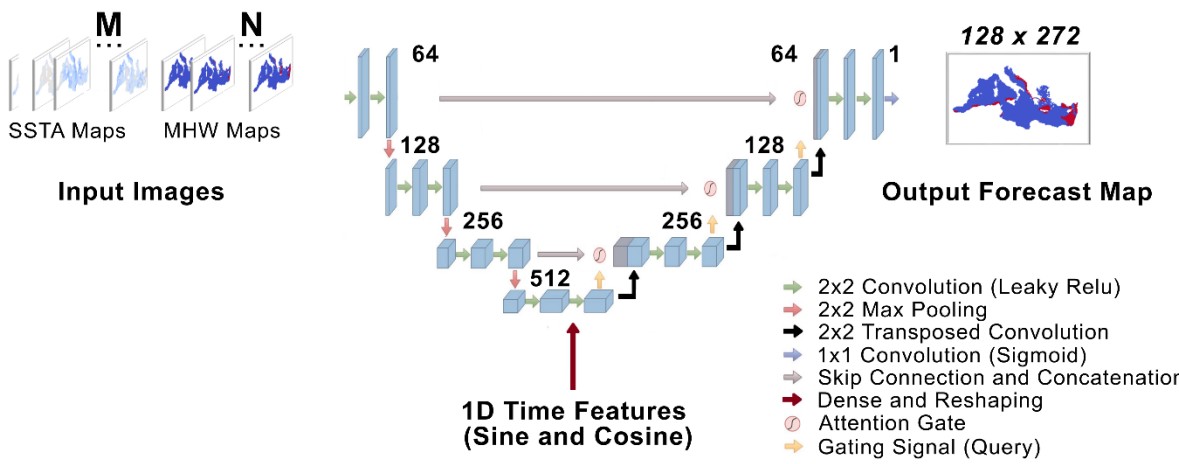

**Figure 2**: *Attention U-net CNN Model architecture to forecast MHW presence/absence maps. N and M are the number of input frames containing spatiotemporal information on MHW presence/absence and SST anomaly, respectively. Figure is adapted by (Ibtehaz and Rahman, 2020).*





path. The contracting path is responsible for extracting features from the input data and progressively reducing the size of the feature maps through downsampling, which, in our case, consists of a series of three Max pooling layers (https://www.tensorflow.org/api_docs/python/tf/keras/layers/MaxPooling2D last accessed online 3/2024).
This process effectively increases the model's ability to perceive broader spatial relationships. As the contracting path progresses, the number of channels in the feature maps is doubled to improve feature capture across different scales. Conversely, the expanding path uses up-sampling operations to restore the feature maps to their original dimensions, consisting of 3 deconvolution layers, using a Conv2D Transpose (https://www.tensorflow.org/api_docs/python/tf/keras/layers/Conv2DTranspose last accessed online 3/2024) for
the decoding path, as shown in Fig. 2.

Moreover, the model incorporates skip connections to integrate both local and global features, enabling the network to utilize information from various scales simultaneously. Attention gates are placed before concatenating the skip connections, automatically learning to focus on target structures of varying shapes and
sizes (Oktay et al., 2018). This design is expected to improve the model's prediction accuracy while maintaining minimal computational overhead (Oktay et al., 2018). The architecture also includes 2D Convolutional layers with 2x2 kernels between all layers. The number of neurons in each layer follows an exponential pattern with a base of 2, increasing in the contracting path and decreasing in the expanding path. The sine/cosine time features from Eq. 1 are incorporated at the bottom of the encoding path, after passing through dense and reshaping layers
to obtain the same spatial dimension as the layer with which are concatenated.

Due to its versatility, this method has been previously used in studies of image segmentation, pattern identification (Oktay et al., 2018; Srivastava et al., 2014) spatiotemporal forecasting (Jacques-Dumas et al., 2022), and downscaling at higher resolutions with minimal computational cost (Ruti et al., 2016) and has been
shown to significantly enhance the accuracy of forecasts. The goal of this network is to determine the probability of each gridpoint being classified as either MHW-present (1) or MHW-absent (0) for predictions of 3 or 7 days ahead, thus forecasting the spatiotemporal probability of MHW occurrence.

## 2.3.2 U-net CNN model Training

A common practice in a neural network approach is the splitting of a dataset into the training, testing and
validation subsets. Here, the validation dataset consists of the early years from 1982 to mid-1986, the training dataset comprises the years from 1986 to 2013 and the final years selected for testing and validation of the model span from 2013 to 2017. Following the split, each of the three subsets (train, test and validation) undergoes random internal shuffling to reduce memorization effects and increase the robustness of the forecasting tool. The tendency for a model to memorize rather than learn meaningful patterns -known as overfitting- is a critical
challenge in NN training, as it leads to excessive tuning to the training data, and causes the model to capture noise and irrelevant details, ultimately compromising its ability to generalize effectively to new, unseen data. To further reduce overfitting effects, we employ a "dropout" approach, which involves random deactivation of a specified number of nodes, set to 30% here, during each training step (Srivastava et al., 2014). An early stopping/best saving checkpoint mechanism is also implemented, whereby the training process halts if there is no





improvement for a predetermined number of epochs and a selected validation metric (i.e., mean squared error, accuracy, recall, f1-score etc.).

Throughout the training of our model, we employ the Adaptive Moment Estimation (Adam) optimizer (Kingma and Ba, 2014), an optimization algorithm, which is based on two gradient descent methodologies, with a batch size of 4 and an initial learning rate of 0.001, set up to reduce by half on a plateau of 10 epochs, with a minimum
learning rate of $10^{-4}$ and maximum number of 100 epochs. Evaluated across all the training samples, the Adam optimizer was used here to minimize the loss function, a key parameter of the Attention U-net CNN, which measures the discrepancy between predicted and actual values. The model's ability to successfully perform a given task is determined by the choice of the loss function and the effective reduction of prediction errors. Given the binary form of the MHW presence (1) and absence (0) fields, the use of a binary cross-entropy loss function
is a common choice (Jacques-Dumas et al., 2022). In the case of extreme events and imbalanced datasets, where one    of    the    two    classes    is    underrepresented,    the    focal    binary    crossentropy    is    preferred (https://www.tensorflow.org/api_docs/python/tf/keras/losses/BinaryFocalCrossentropy    last    accessed    online 3/2024) (Lin et al., 2020). This function further improves the effects of the standard binary cross-entropy by integrating two additional parameters, designed to reduce the influence of correctly classified samples and
emphasize the importance of misclassified ones. We use this loss function, which in equation form reads as:

$$Focal(p_t) = -\sum^{N \cdot M} \alpha_t (1-p_t)^\gamma \log(p_t) \ where \ p_t = \begin{cases} p & , y=1 \\ 1-p & , otherwise \end{cases}, \tag{Eq. 2}$$

where y is the ground truth class, p ∈ [0,1] is the predicted probability and $a_t$ are the tuning parameters, which are selected as 0.25 and 2, following Nguyen and Thai (2023). This adjustment leads to elevated loss values when misclassifications occur, steering the training process toward lower local minima of the loss function. Due to the
non-linear nature of the patterns the neural network is trained on and despite the attempts to minimize the loss function, its convergence often leads to near local minima, impeding its performance. To improve the model's performance and ensure that minima values of the loss function maintain satisfactory accuracy levels in their prediction, several iterations with varying parametric choices are conducted on test cases (see Results Sect. 3.2, Table 1).


An additional aspect of neural network architecture is the selection of activation functions, which are applied to the outputs of each intermediate hidden layer. These functions have a key role in determining the operations applied to the input neurons and thereby the model's ability to generate an output (Sharma et al., 2020). In this study we overcome the limitations associated with the use of a standard Rectified Linear Unit (ReLu) activation
function in handling negative input values, by using a Leaky ReLU version in all the intermediate layers (Maas et al., 2013). The Leaky ReLU function allows a small, non-zero gradient for negative inputs, effectively mitigating the vanishing ReLU problem, where neurons become inactive during training. This effect proves advantageous for non-linear prediction tasks, enabling the model to capture a broader range of input variations. The output (final) layer is equipped with a sigmoid activation function, which is appropriate for binary classification tasks, as
it produces values within the range of [0,1]. These values express the probability for a grid point being affected by a MHW, with the classification threshold determined by the specific characteristics of the physical problem in



each instance. The process by which we select the appropriate classification threshold is further discussed in Sect. 3.1.

The U-net CNN model described here was implemented using the Keras API, using Tensorflow 2.9.2 in Python. The architecture outlined in Fig. 2 required 32 million parameters to train, and it run in parallel on 8 Nvidia A100 GPUs with a memory of 40 GB each, in one node of the p4d.24xlarge Amazon EC2 server. Each test case required approximately 4 hours to complete the training of 100 epochs, and the inference speed required seconds to calculate forecasts for each sample on the same server. We note that once the computationally expensive

training ends, the model can be deployed to any system to generate forecasts, with minimal computational overhead, given the appropriate input data. For instance, the inference time for the entire testing dataset, consisting of 1300 daily forecasts, required approximately 60 seconds on the aforementioned hardware configuration.

**2.4 Evaluation Metrics**

The model's performance is evaluated on a testing dataset consisting of samples that were obscured by the U-net CNN during its training phase. At this stage, we assess the model's ability to accurately predict unseen data and determine the optimal probability threshold, above which each grid point is categorized as a MHW-affected case. Throughout the training and validation process, standard metrics such as Recall, accuracy and the selected loss

function are calculated during each epoch for the training dataset and at the end of each epoch for the validation dataset (https://www.tensorflow.org/api_docs/python/tf/keras/metrics_last_accessed_online_3/2024). The primary metric to evaluate the prediction skill of the Attention U-net CNN on future MHW occurrence is the True Positive Rate (TPR). This rate assesses the proportion of accurately predicted MHW occurrences relative to the total number of actual occurrences (Sun et al., 2023). The formula for calculating TPR is given by:

$$TPR = \frac{TP}{TP + FN},$$

285                                                                                                                     (Eq. 3)

where TP represents True Positive predictions and FN denotes False Negative ones. In other words, the TPR metric quantifies the model's ability to accurately detect true MHW occurrence during testing. At the end of each epoch during the training process, validation Recall, a metric similar to TPR, is computed on the validation subset and serves as an early-stopping mechanism, a technique useful for both computational efficiency and preventing

overfitting. In contrast, the True Negative Rate (TNR) evaluates the model's ability to accurately predict the negative class labels, specifically the absence of a MHW, relative to the total number of non-MHW occurrences. This metric reads as:

$$TNR = \frac{TN}{TN + FP},$$

                                                                                                                        (Eq. 4)

where TN represents the True Negatives and FP the false positives. Combining these two rates, the Forecast

Accuracy Rate (FAR) composite metric can be used to assess the model's overall forecasting ability. Unlike the TPR and TNR which focus on a single class, the FAR considers both the correct and incorrect MHW predictions across all classes. In formula form, FAR is defined as:



$$FAR = \frac{TN + TP}{TN + FP + TP + FN},$$

(Eq. 5)

providing a percentage-wise estimation of forecast accuracy.


To obtain an overall TPR, TNR and FAR as single numerical indicators for each sample in the testing dataset we separately average the metrics defined in (3), (4) and (5) over the Mediterranean Sea domain. This approach provides a single, spatially independent numerical value for each error metric corresponding to each sample in the testing dataset. The final TPR, TNR and FAR for each forecast horizon are determined by averaging each of

these metrics across all test samples.

## 2.5 Persistence Benchmark

A standard approach to evaluating the predictive skill of our ML model is to compare its output forecast with a climatological baseline derived from a Persistence benchmark model, which assumes that MHW presence or

absence, within the next 3 or 7 days, remains constant throughout the forecast period. Specifically, the Persistence benchmark uses MHW presence/absence fields from 3 or 7 days prior to an event as both the input and the forecast for the target MHW conditions. By operating under the assumption that recent MHW conditions persist into the forecast period, the Persistence model provides a baseline performance level against which we assess whether the U-net CNN model adds predictive value beyond simple temporal Persistence. Despite the

simplicity of this assumption, using the Persistence benchmark model as a reference dataset is a meaningful approach for short-term forecasting (Parasyris et al., 2022), given the relatively slow changes in SST, which lead to minimal variations in MHW presence on most consecutive days. Improvements demonstrated by the output forecast of a neural network model that surpass the basic Persistence model are essential for accurately forecasting the onset and dissipation of MHWs.


## 3 Results

### 3.1 Assessing the probability of MHW occurrence

The output probabilities of MHW occurrence are converted into binary classification masks as an initial step to assess the forecast ability of our ML model. In particular, the U-net CNN generates maps where each pixel value

ranges between 0 and 1, representing the probability of each grid point being a negative (MHW-absent) or positive (MHW-present) class (referred to as forecast probability). A range of forecast probability thresholds (hereafter, threshold) are then evaluated from 0.05 to 0.9, in increments of 0.05, as a trial-and-error technique, to find the one which maximizes a specific accuracy metric (TPR, TNR, FAR). A prediction based on each threshold is subsequently generated for the entire testing dataset and compared with the true MHW occurrences

originally identified using the climate model output. Following the work of Sun et al. (2023), we first compute the spatially averaged TPR and TNR separately for each threshold, and further average their sum for all samples within the test set. The optimal thresholds for the 3-day (Fig. 3a) and 7-day (Fig. 3b) forecast scenarios are then determined by maximizing this Combined Mean (CM) of TPR and TNR, which is formulated as:






$$CM = \frac{TPR + TNR}{2},$$ (Eq. 6)

In both scenarios, increasing the threshold results in a higher TNR and lower TPR, as more predictions are classified as MHWs absences, the accuracy of which is thereby improved. The intersection point in each forecast scenario (Fig. 3, red star) represents an equilibrium between sensitivity and specificity in classification terms.

This point maximizes the CM (Fig. 3, green lines), optimizing predictive performance of both the positive and negative MHW occurrences. Specifically, the optimal threshold for the 3-day forecast is identified at 0.45, yielding a maximum CM of 0.89 (Fig. 3a), whereas the 7-day forecast achieves a CM of 0.77, at an optimal threshold of 0.4 (Fig. 3b). At a high threshold of 0.85, both scenarios exhibit a TNR close to 1, indicating no false negatives, as most predictions are classified as negative. However, at this threshold, the respective TPR is 0.48

for the 3-day forecast and 0 for the 7-day forecast (Fig. 3, blue lines). This suggests that for the 3-day forecast,

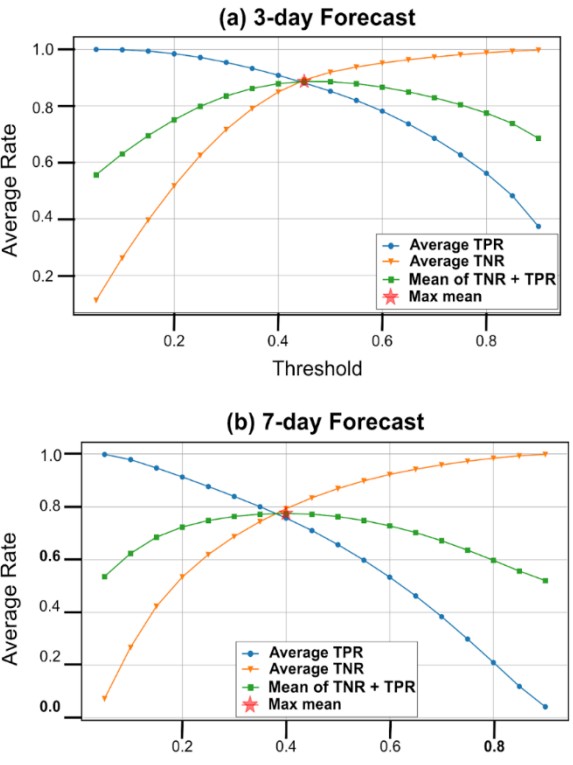

**Figure 3***: Threshold selection for the a) 3-day and (b) 7-day forecast scenario based on the maximization of the combined mean (green line) of TPR (blue line) and TNR (orange line). The threshold increments range from 0.05 to 0.9 and the maximal point is indicated by a red star.*

half of the MHWs were correctly classified despite the excessively high threshold. In contrast, the 7-day forecast shows a decline in performance, as TPR is approximately zero (Fig. 3b).



In the context of optimizing the threshold selection, TPR (Eq. 3) and TNR (Eq. 4) are prioritized over FAR (Eq. 5), as the primary goal is to accurately predict MHW occurrences (True Positives) rather than minimizing false alarms (FP, FN). Given that FAR includes all grid points in the domain (128 x 272), it relies on a larger denominator, resulting in lower overall values. This causes the FAR curve to shift toward higher threshold values, consequently reducing the number of True Positives, which is suboptimal. Nevertheless, FAR is useful as a cumulative metric, and is later used to assess the spatial overall performance of the optimal model configuration.

## 3.2 Sensitivity Analysis on Input Variables

To further optimize the configuration of our U-net CNN, we perform a sensitivity analysis on the input variables employed in training the model. Specifically, 9 distinct experiments are examined for both the 3-day and 7-day forecast scenarios, in which we varied the number of SSTA days and MHW presence/absence window preceding the target prediction date. The number of experiments was primarily limited by the memory constraints of our computational setup during training. Additionally, we prioritized simplicity to ensure minimal requirements during forecasting, enhancing the tool's practicality as a viable alternative for MHW prediction. For each experiment we evaluate the corresponding TPR, TNR, CM and FAR metrics and determine the best-performing setup of our model based on the maximum CM. The decision to prioritize CM (over FAR) reflects the greater influence of TPR on CM, aligning with our primary objective of accurately predicting positive MHW occurrences.

**Table 1**: *Sensitivity experiments on the input variables of the U-net CNN model and the associated prediction metrics for the 3-day forecast scenario. The input variables configuration for each experiment is indicated with the number of preceding timesteps of SSTA (N) and the number of preceding timesteps (M) of MHW presence/absence maps. The highest evaluation metrics are highlighted in bold, for clarity. The TPR, TNR, CM and FAR represent True Positive Rate, True Negative Rate, Combined Mean and Forecast Accuracy Rate, respectively.*

| Experiment | M | N | TPR | TNR | CM | FAR |
|---|---|---|---|---|---|---|
| 1 | 5 | 0 | 0.785 | 0.726 | 0.756 | 0.783 |
| 2 | 5 | 2 | 0.844 | 0.898 | 0.871 | **0.918** |
| 3 | 5 | 4 | 0.864 | 0.890 | 0.880 | **0.918** |
| 4 | 10 | 0 | 0.758 | 0.759 | 0.758 | 0.786 |
| 5 | 10 | 2 | **0.881** | 0.891 | **0.886** | 0.900 |
| 6 | 10 | 4 | 0.858 | **0.905** | 0.882 | 0.902 |
| 7 | 14 | 2 | 0.865 | 0.893 | 0.879 | 0.917 |
| 8 | 14 | 4 | 0.867 | 0.850 | 0.859 | 0.914 |
| 9 | Persistence | | 0.844 | 0.890 | 0.867 | 0.881 |



In the case of the 3-day forecast scenario, experiment 5 achieves the highest CM, incorporating M=10 preceding timesteps of SSTA and N=2 preceding timesteps of MHW presence/absence maps as input variables, with a total
of 31.926.629 parameters trained (Table 1). Although this experiment demonstrates the highest TPR (0.881) and the second highest TNR (0.891) among all experiments, it does not yield the maximum FAR, which is observed in experiments 3 and 2. Overall, the TPR and TNR variations across most of the sensitivity tests in the 3-day forecast scenario, remain within a 5% range, indicating a relative stability of the model's performance and robustness to small changes in input variables. However, a marked deterioration in performance is observed in
experiments 1 and 4, with a TPR declining by up to 16%, compared to other experiments. This is due to the SSTA being the sole input variable (N=0) of this configuration.

In comparison, the 7-day forecast produces slightly reduced metrics, indicating decreased accuracy over the extended forecast period (Table 2). In particular, the highest CM is identified in experiment 7, which incorporates
M=14 preceding timesteps of SSTA and N=2 preceding timesteps of MHW presence/absence maps as input variables. While this experiment demonstrates a balanced performance, achieving a TPR of 0.757 and the second highest TNR (0.792), experiment 9 (the Persistence benchmark) exhibits the highest TNR (0.872) and a significantly lower TPR (0.653).

**Table 2** *As in Table 1 but for the 7-day forecast scenario.*

| Experiment | M | N | TPR | TNR | CM | FAR |
|---|---|---|---|---|---|---|
| 1 | 5 | 0 | 0.753 | 0.688 | 0.721 | 0.718 |
| 2 | 5 | 2 | 0.704 | 0.812 | 0.758 | 0.823 |
| 3 | 5 | 4 | 0.761 | 0.735 | 0.748 | 0.778 |
| 4 | 10 | 0 | 0.774 | 0.648 | 0.711 | 0.716 |
| 5 | 10 | 2 | 0.735 | 0.794 | 0.764 | 0.801 |
| 6 | 10 | 4 | 0.7565 | 0.759 | 0.757 | 0.788 |
| 7 | 14 | 2 | **0.757** | 0.792 | **0.775** | 0.802 |
| 8 | 14 | 4 | 0.756 | 0.754 | 0.755 | 0.780 |
| 9 | Persistence | | 0.653 | **0.872** | 0.763 | **0.829** |

Compared to the 3-day forecast, the TPR and TNR variations across the sensitivity tests of the 7-day forecast, reach up to 14% (Table 2). For instance, TPR ranges from 0.653 in experiment 9 to 0.774 in experiment 4, while TNR ranges from 0.648 in experiment 4 to 0.872 in experiment 9. When excluding cases that lack MHW
presence/absence maps as an input variable (N=0), the results show the same model stability demonstrated in Table 1, with TPR and TNR variations confined within a 5% range. Indeed, a marked deterioration in performance is observed in experiments 1 and 4, with TPR values ranging between 0.753-0.774 and TNR values




between 0.688-0.648, respectively. This indicates that the model's ability to effectively leverage patterns from past events is limited when information about MHW presence/absence maps is excluded.


### 3.3 Performance of optimized U-net CNN configuration

In the following, we assess the forecasting ability of our U-net CNN model based on the best performing configurations of each forecast horizon. Specifically, we assess the predictive capability of experiment 5 for the 3-day forecast scenario and experiment 7 for the 7-day forecast scenario, which were determined to yield the 420 highest performance metrics.

#### 3.3.1 Forecast Rates

The performance of the binary MHW classification is evaluated across the entire testing dataset (mid 2013 to 2017), irrespective of temporal or spatial variations, using a confusion matrix. This tool quantifies the percentage 425 of correctly and incorrectly classified grid points within the MHW spatial domain throughout all the samples of each forecast scenario, providing a comprehensive evaluation of the model's predictive accuracy and robustness. The classification metrics used include the TPR and TNR as well as the False Positive Rate (FPR) for negative cases misclassified as positive and the False Negative Rate (FNR) for positive cases misclassified as negative.

The model achieves high accuracy in correctly classifying both the MHW-affected (True Positive) and the non-MHW grid points (True Negative) in the 3-day forecast scenario, with success rates of 88.1% and 89.1%,

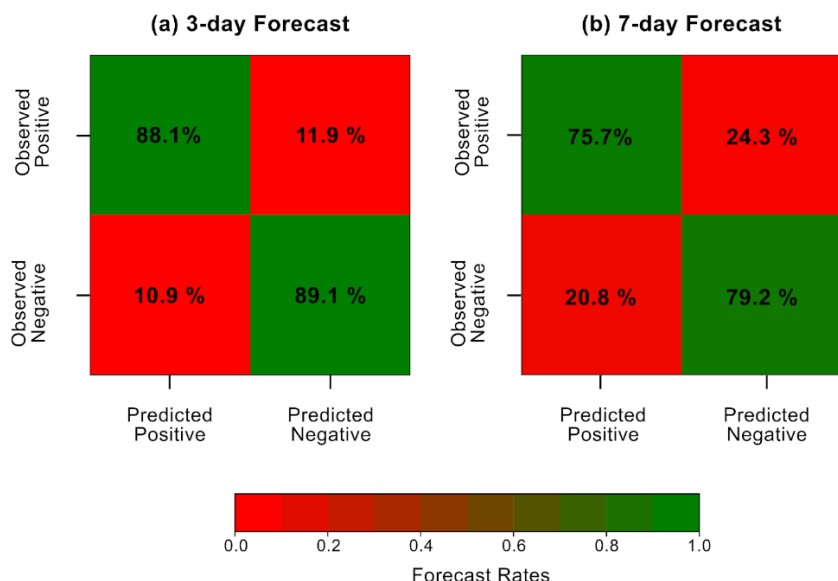

**Figure 4**: *Confusion plot for the (a) 3-day and (ii) 7-day forecast, showing the TPR (Observed Positive-Predicted Positive), TNR (Observed Negative-Predicted Negative), FPR (Observed Negative-Predicted Positive) and FNR (Observed Positive-Predicted Negative). Forecast rates are denoted in percentages within the boxes, taking into account all the samples of the testing dataset spanning mid-2013 to 2017.*



respectively (Fig. 4a). Incorrect predictions account for only up to 12% of the samples. In comparison, relatively lower rates of TPR (75.7%) and TNR (79.2%) are seen in the 7-day forecast (Fig. 4b), likely due to the reduced autocorrelation of the 7-day lagged input maps, compared to the 3-day case, with the proportion of misclassified
cases increasing to 24%.

To complement the analysis of aggregate metrics, we further examine the predictive accuracy of our U-net CNN on a sample-by-sample basis within the testing dataset, spanning mid-2013 to 2017 (Fig. 5). Specifically, we compare the total number of predicted positive MHW grid points to the observed positives for each sample. By
examining the alignment of data points with the line of parity (Y=X), we assess not only the overall predictive skill but also patterns of systematic overprediction or underprediction within the dataset. This approach enables us to identify deviations from perfect agreement and provides insights into potential biases in the forecasts. For the 3-day forecast, a strong agreement is observed between the predicted and observed MHW occurrences, as 99.31% of the points fall within a tolerance of 3,500 grid points from the Y = X line (Fig. 5a). This tolerance
threshold was empirically determined to optimize performance in the 3-day forecast scenario, enabling meaningful comparisons with the 7-day forecast. Given that in geospatial analysis it is common to use a threshold

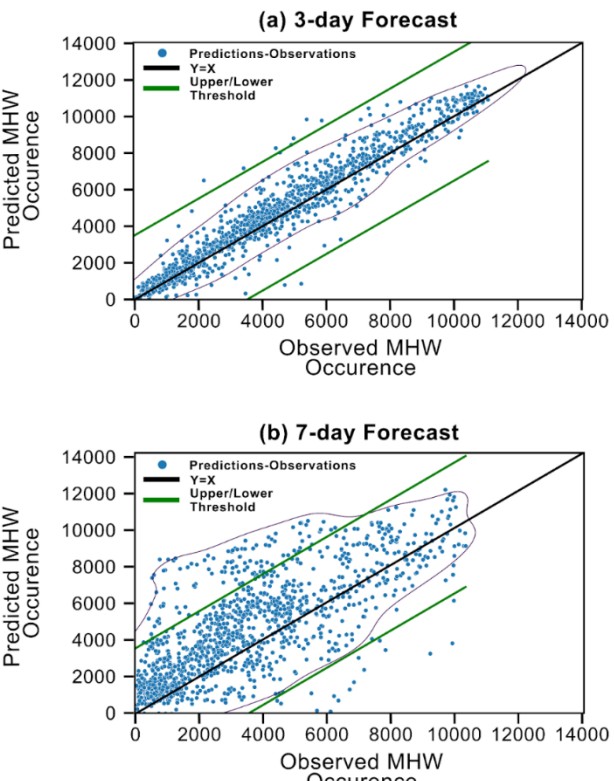

**Figure 5**: *Scatter plots of the total number of observed (x-axis) vs predicted (y-axis) points classified as MHW present (blue dots) per sample, for the a) 3-day and b) 7-day forecast throughout the entire testing dataset spanning mid-2013 to 2017. The upper and lower thresholds of 3500 points (green lines) are introduced for comparison purposes (see text).*




based on a percentage of the total dataset to ensure stability and scalability (Xu et al., 2024), we chose a tolerance that represents approximately 10% of the total grid points in each map. In contrast, the 7-day forecast scenario shows reduced alignment, with 86.68% of samples meeting the same tolerance threshold. Furthermore, most

points lie above the Y = X line, indicating a potential overestimation of MHW presence in this scenario (Fig. 5b). These results highlight the ML model's diminished predictive reliability at extended forecast horizons, consistent with the inherent trade-offs in balancing sensitivity and specificity.

### 3.3.2 Comparison with the Persistence Benchmark

The final phase of evaluating the U-Net CNN configuration focuses on comparing its performance to the Persistence benchmark model, which predicts MHW presence/absence based on lagged correlations over 3-day and 7-day intervals (Fig. 6). The Persistence model predicts MHW occurrence based on past observations,

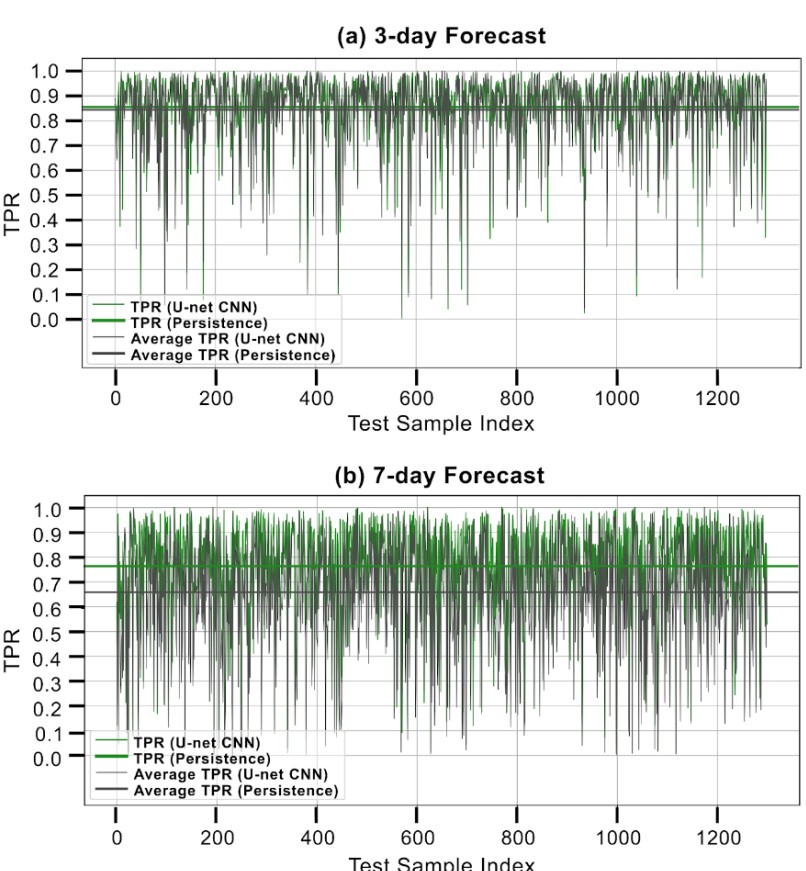

**Figure 6:** *TPR values of each sample across the entire 3.5-year test dataset spanning mid-2013 to 2017 for the best-performing experiment and Persistence benchmark of the a) 3-day and b) 7-day forecast. The TPR values of each U-net model are indicated in green lines whereas TPR of the Persistence model is shown in black. The averaged value across the entire 3.5-year test dataset is also shown with the same colors as a bold line, clearly indicating which method is better overall.*



assuming that the most recent conditions are the best predictor of future events. The "noisy" TPR fluctuations observed across the test dataset can arise from rapid changes in the MHW presence/absence maps, which
challenge the predictive stability of both models. This variability is further compounded in the U-Net CNN by the inherent stochasticity in neural network predictions. However, in terms of the average TPR, the U-net CNN consistently outperforms the Persistence benchmark across both forecast scenarios (Fig. 6). For the 3-day forecast, the Persistence model achieves an average TPR of 0.844 (Fig. 6a), which declines to 0.653 in the 7-day forecast (Fig. 6b), reflecting stronger performance in the shorter forecast horizons. In comparison, the reduction
in TPR between the 3-day and 7-day forecast is less pronounced for the U-net CNN, declining from 0.881 to 0.757. This indicates that the U-net CNN exhibits a comparatively higher stability in maintaining predictive performance over longer forecast horizons. Overall, the best-performing experiments of the U-net CNN outperform the Persistence benchmark model, achieving higher values across all evaluation metrics in the 3-day forecast and in some select metrics for the 7-day forecast. This reflects the model's robustness in shorter forecast
horizons and its ability to maintain competitive performance despite the challenges posed by longer forecast horizons.

### 3.3.3 Evaluation of Spatial Prediction Accuracy

This section investigates the regions of the Mediterranean Sea displaying higher and lower susceptibility to
prediction errors in our method. To this end, we examine the spatial distribution of the averaged FAR across all samples of the testing dataset (mid-2014 to 2017) for both forecast scenarios. This metric provides a cumulative

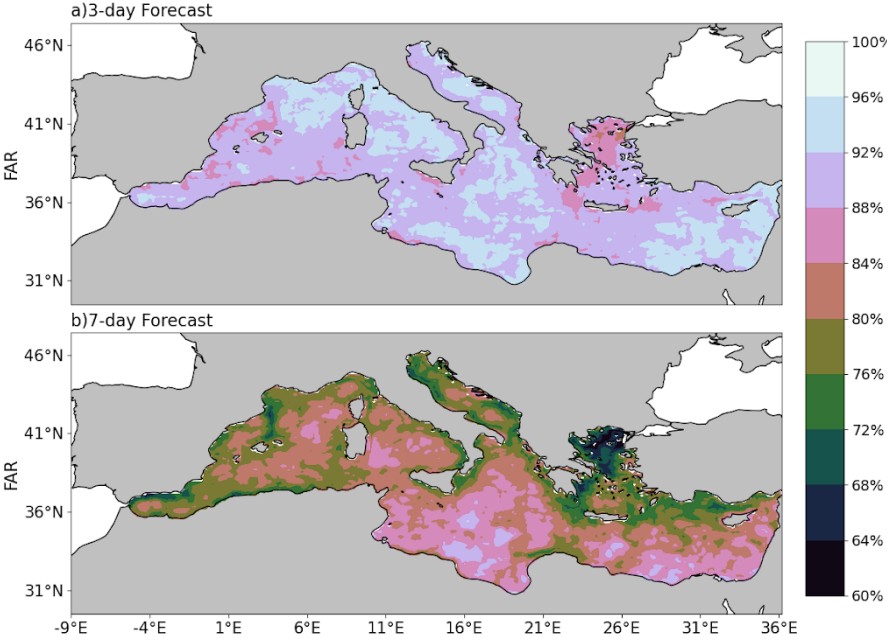

**Figure 7:** *Spatial distribution of the Forecast Accuracy Rate (FAR) in the Mediterranean Sea for the a) 3-day and b) 7-day forecast scenario, averaged across all samples of the testing dataset spanning mid-2013 to 2017.*





assessment of the U-net CNN model's predictive performance, enhancing our understanding of regional prediction reliability, while guiding model refinements to improve forecast precision in identified areas of weakness.

Overall, the spatial distribution of FAR reveals significant differences between the 3-day and 7-day forecast scenarios. In the 3-day forecast, FAR values exceed 90% in the northwest Mediterranean, the Adriatic and Ionian Sea as well as the southeast Mediterranean basin, as opposed to the Balearic Islands, Alboran and Aegean Sea, where slightly lower FAR (80% - 90%) is displayed (Fig. 7a). In the 7-day forecast, FAR values are generally lower across the entire Mediterranean Sea. Specifically, FAR values in the Ionian and Tyrrhenian Sea as well as

the Levantine basin, range between 80% - 92%. In contrast, the Aegean Sea as well as the coastal areas of the northern Mediterranean basin and the Alboran Sea, display FAR values approximately between 60%-70% (Fig. 7b).

## 4 Discussion and Conclusions

This study presents the application of an Attention U-Net CNN to predict the spatiotemporal evolution of MHWs in the Mediterranean Sea. The proposed model integrates an attention mechanism with a standard U-Net architecture, combined with a focal binary cross-entropy loss function, among other key parameters. The model is trained on SST anomalies and information on daily MHW presence/absence between 1982-2017 to predict future MHW occurrence within 3-day and 7-day forecast horizons. Extensive hyper parameter tuning is carried out to

ensure the model's stability and performance within acceptable limits, alongside the implementation of a specific threshold selection technique. The results, presented for both forecast horizons, highlight a decline in forecasting skill as the prediction horizon increases, with our U-net CNN consistently outperforming the Persistence benchmark model.

For the optimal thresholding technique of MHW probability, we prioritize the accurate detection of true MHW instances (TPR) over minimizing the number of non-MHW instances (TNR). Specifically, we focus on the maximization of the combined mean of TPR and TNR (CM), to ensure a balance between sensitivity and specificity (Sun et al., 2023). As the threshold increases, a trade-off emerges between the TPR and TNR (green line, Fig. 3), whose curves form a concave shape with a distinct maximum point. In cases where this maximum is

not well-defined, an optimal threshold can be determined either empirically (Fawcett, 2006), based on the intersection of TPR and TNR (Fig. 3), or by maximizing the FAR metric (Sun et al., 2023).

The optimal configuration of the U-net CNN is then determined through sensitivity analysis, assessing the impact of various input variable combinations on model performance. The best-performing configuration incorporates

M=10 preceding SSTA timesteps and N=2 preceding timesteps of MHW presence/absence for the 3-day forecast (Table 1) and M=10 and N=4 for the 7-day forecast (Table 2). In both forecast scenarios, configurations relying solely on SSTA data demonstrate the weakest performance, highlighting the importance of integrating both temperature anomalies and prior MHW occurrences to achieve more reliable forecasting of these events. Particularly for the 7-day forecast the variations in the CM and FAR metrics across the different experiments are

less pronounced when excluding experiments with a single input variable. In line with the findings of Sun et al.



(2023), this sensitivity analysis aims to optimize the model's performance on unseen data, while maintaining simplicity for future end-users and balancing model complexity and generalization.

However, the assessment of the U-net CNN's forecast ability reveals distinct differences in predictive accuracy between the short-term (3-day) and longer-term (7-day) forecast horizon. While the 3-day forecast achieves the highest average TPR and TNR metrics overall, its discrepancy relative to the Persistence benchmark is modest, potentially due to the high autocorrelation of the SSTA in both the benchmark and the 3-day forecast (Fig. 6a). In contrast the 7-day forecast exhibits lower TNR and TPR values, with the performance gap between the benchmark and the U-net CNN being more pronounced (Fig. 6b). Thus, our results indicate an improved accuracy in shorter forecast horizons, in agreement with Bonino et al. (2023b). Despite demonstrating a decline in TPR between the two forecast scenarios, the strength of the U-net CNN lies in maintaining a higher TPR value than the Persistence benchmark in both forecast scenarios, suggesting an improved ability to predict MHWs across different temporal scales.

Based on the FAR metric, an improved forecasting performance (high FAR values) of the U-net CNN is revealed in the northwest Mediterranean basin, the Ionian Sea as well as the Levantine basin, as opposed to the lower FAR values observed in the Aegean and Alboran Sea, the Balearic Islands and northern coastal areas (7-day forecast). In the case of the Aegean Sea, the low FAR values may reflect challenges in detecting MHWs due to rapid SST fluctuations influenced by the Black Sea. These fluctuations can lead to swift onset and dissipation of MHWs (Mavropoulou et al., 2016), complicating their detection by ML methods. Additionally, the downsampling applied to address the high computational demands of our method (see Sect. 2.2) resulted in a reduced spatial resolution of the data. This reduction likely affected the model's performance in regions with complex topography, such as the Aegean Sea and coastal areas. This is also observed in Bonino et al. (2023b) where the authors also report a reduced forecast ability of their neural network model around the Adriatic Sea, Balearic islands, the Alboran and Aegean Sea.

It is important to note here that the results of this study are based on the "straight split" methodology, where the early years of the dataset are used for training and the final years for testing. Given that ML models trained on recent data often achieve higher accuracy, due to the influence from recent climate patterns (recency effect) (Lam et al., 2023), we have also explored the impact of reversing the training and testing dataset order. In particular, a sensitivity test that we define as the "opposite split" was carried out, using the years 2013–2017 for training and 1982–1986 for testing, to assess whether our model's predictive skill is dataset-dependent or driven by climate change-induced temperature. For the 7-day forecast horizon, we find improved TNR and TPR metrics when more recent data are inserted in the training dataset. This aligns with the findings of Lam et al. (2023), where recent climate trends, including the increased frequency of MHWs, can enhance model effectiveness by mitigating issues of data scarcity and imbalance in the training datasets. However, for the 3-day forecast, we find a deterioration of both metrics, compared to the outputs of the "straight split" methodology (see Supplementary Material Table S1). This may be due to the longer duration of recent MHWs, which are less frequent in earlier years (Oliver et al., 2018), and thus may be poorly represented in the training dataset. Given that training a neural





network with historical data to forecast information in the future reflects a more realistic approach, in this study
we adopted the "straight split" methodology.

Overall, the proposed U-net CNN model offers a computationally efficient alternative to traditional regional
forecasting models for predicting MHWs in the Mediterranean Sea. Once trained, our approach maintains high
spatiotemporal resolution while requiring minimal computational resources. The results indicate that the model
performs better in the 3-day forecast compared to the 7-day forecast across all evaluation metrics and relative to
the Persistence benchmark. This outcome is expected, given the higher autocorrelation of SSTA over shorter
timeframes, consistent with other studies (Sun et al., 2023; Taylor and Feng, 2022). The 7-day forecast, however,
holds greater practical value, as earlier predictions of extreme events allow for more effective mitigation
strategies. At the same time, the 7-day forecast is a more challenging task, as shown by the lower Persistence
benchmark values over longer time horizons (see Tables 1 and 2). In comparison, the 3-day forecast achieves
higher overall metrics due to the shorter forecasting window and the inherently higher Persistence benchmark
values. Nevertheless, the U-Net model still outperforms the benchmark in the 3-day case, although by a smaller
margin than in the 7-day forecast.


While incorporating atmospheric variables into the training process could presumably enhance the model's ability
to capture broader climatic influences on MHW occurrence, the inclusion of additional variables, such as air
temperature, did not improve forecast performance for the specific optimal configuration of input timesteps.
Moreover, the selection of a limited set of training variables contributes to the model's simplicity and replicability
for both the training and prediction phases.

As global warming accelerates, the increasing frequency and severity of MHWs pose significant challenges to
marine ecosystems. Efficient and timely forecasting of MHWs is essential for effective marine management,
enabling governments, industries, and coastal communities to take proactive measures, such as imposing
temporary fishing bans, enhancing monitoring of vulnerable species, establishing marine protected areas, and
launching public awareness campaigns to promote sustainable practices. ML-based approaches show promise for
improving predictions of these events, particularly through architectures like the Attention U-Net CNN employed
in this study, which reconstruct spatially distributed variables and generate high resolution predictions, within
seconds to minutes, depending on available computational resources. This rapid forecasting ability is particularly
advantageous for short-term predictions, as climate models typically require days of runtime and data
assimilation to achieve comparable accuracy in operational settings (Coppini et al., 2023).

However, ML-based approaches face limitations that need to be addressed to ensure forecasts remain reliable and
robust. For instance, while CNNs can process large sets of unstructured data arrays and produce reliable spatial
predictions, they require significant computational resources, due to extensive data handling and intricate
computational procedures. Thus, a critical consideration in the application of ML techniques is balancing
predictive accuracy with computational demands. Additionally, many ML models violate fundamental physical
principles, reducing their precision and long-term reliability (Chen et al., 2023). To mitigate this, Desai and
Strachan (2021) proposed the integration of physical laws into a ML framework, where a genetic algorithm was




coupled with a neural network model, while Zanetta et al. (2023) suggested using analytical equations in DL-based post-processing to enhance prediction reliability. These approaches, though promising, remain in early stages. Forecasting MHW intensity and duration, particularly for long-term predictions, is further hindered by error propagation. Future research could involve training ML models on observed and/or remotely sensed data, which were not used here as they come with their own set of limitations (Abdelmajeed and Juszczak, 2024).

Moreover, while ML excels at short-term forecasts, traditional climate models perform better since they depend heavily on atmospheric forecasts to capture the dynamics influencing ocean behavior (Bonino et al., 2023a). Therefore, a hybrid approach that combines the strengths of ML and the long-term reliability and physical consistency of traditional models could be a powerful tool for forecasting MHWs. Future efforts should focus on refining the current limitations of data-driven methodologies and enhance predictive capabilities.


**Data availability:** Data used are cited when firstly introduced and are available from corresponding authors upon request.

**Author Contributions.** Conceptualization: **A.P.** Data curation: **A.P., S.D.** Formal analysis: **A.P., S.D., V.M.** Funding acquisition: **S.D.** Investigation: **A.P., S.D.** Methodology: **A.P., S.D., V.M.** Project administration: **S.D.**
Resources: **A.P., S.D., N.K.** Software: **A.P.** Supervision: **S.D.** Validation: **A.P., V.M.** Visualization: **A.P., S.D., V.M.** Writing – original draft preparation: **A.P., S.D.** Writing – review & editing: **S.D., V.M., A.P.**

**Competing interests.** The authors declare that they have no conflict of interest.

**Acknowledgements:** Computational time is granted by Meluxina under ID u100862 and AWS EC2 server under GRNET-EDYTE Amazon grant.

**Financial support.** Hellenic Foundation for Research and Innovation (H.F.R.I.) under the 3rd Call of "Research Projects to Support Post-Doctoral Researchers" scheme (Project Number 07077, acronym TExMed)

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
