# Peer review of "Marine Heatwaves in the Mediterranean Sea: A Convolutional Neural Network study for extreme event prediction"

_EGUsphere, 2024_

## Author Response (AR1)

**Marine Heatwaves in the Mediterranean Sea: A Convolutional Neural**

**Network study for extreme event prediction**

Response to Reviewers

We thank the reviewers for their thoughtful and constructive feedback. Below, we provide detailed responses to their final comments, highlighted in blue, while retaining the reviewers' original text in black for reference. The line numbers mentioned correspond to the revised manuscript (clean version, without track changes)

**Reviewer #1**

The authors applied a relatively new kind of artificial intelligence to examine marine heat waves (MHWs) in the Mediterranean Sea. They highlight both the strengths (ability to issue reasonable forecasts) and weaknesses (e.g. computational intensity) of the method. There are some minor revisions and comments to address, but I recommend this manuscript for publication.

We thank the reviewer for their kind comments and devoting time to reading our manuscript. We believe that addressing their feedback has significantly strengthened the overall quality of the paper.

One significant addition, preferably in the paragraph starting on line 137, would be to note the important fact that MHW is defined based on a 30-year climatological mean. With most SST data sets now going back to the 1980s, the selection of a 30-year window can alter the results of an experiment like this, especially since many regions have statistically significant different temperatures now compared to 30-40 years ago. Which 30 year window was used for the identification of MHWs, or is it based on the window 1982-2017 like the SST anomalies?

We thank the reviewer for their suggestion. We have now clarified in the manuscript that the years 1982-2014 were used as the baseline climatology, spanning 32 years (line 134).

It appears that the AI model used is fairly useful for short-term forecasts. Could you please note the merits of this relative to other AI methods used in climatology?

We thank the reviewer for their comment. In climatological prediction exercises, AI models require careful consideration of the forecasting horizon in relation to input and output frequency. For long-term climate predictions, several decades of input data and coarser time steps are generally required to achieve high accuracy. In our study, however, the need for short-term forecasts and the availability of data led us to use daily maps as inputs. This enabled us to achieve high TPR/TNR values. Given the inherent spatiotemporal characteristics of our dataset, the CNN-U-net architecture we used, which effectively captures both spatial and temporal dependencies, was deemed one of the few suitable ML approaches for this application. Furthermore, we introduced modifications in our CNN U-net model, such as the tailored loss functions and the attention mechanism, in order to distinguish our approach from the standard U-net methodologies.

Line 30: Generally, we use $^{o}C$/decade or $^{o}C$/century.

We thank the reviewer for their suggestion. However, as this value is derived from a previously published study, we have maintained the original format, with a slight revision to the sentence.: "*Since the 1980s,*

*the Mediterranean Sea has experienced a mean SST increase of approximately 0.041°C/year, which is twice the global average (Pisano et al., 2022), lines 29-30.*

Line 50: "ML" needs to be defined here.

We appreciate the suggestion. The term has now been defined upon its first occurrence in the text for clarity: line (50).

Line 87: "Artificial Neural Network" needs no capitalization

Thank you for noting this. We have now corrected it throughout the text.

Fig. 1: Ierapetra Gyre is labeled with both an arrow and one of the red rings

Thank you for your observation. This has now been corrected.

Line 155: "SSTA" needs to be defined here.

We appreciate your suggestion. "SSTA" has now been explicitly defined upon its first occurrence in the text for clarity: line (102). Additionally, SST abbreviation has been defined on first occurrence on line 29.

Fig. 2: is the domain 128x272 the size of the data region in terms of data matrix size?

Yes, exactly. We have added an explanation in the caption of Figure 2 to make this clearer. Now, it reads: "**Figure 2***: Attention U-net CNN Model architecture to forecast MHW presence/absence maps. N and M are the number of input frames containing spatiotemporal information on MHW presence/absence and SST anomaly, respectively. Each map corresponds to daily frequency of input data and has a matrix size of 128 x 272. U-net figure is adapted by Ibtehaz and Rahman (2020)"*

Eq. 2: y and γ are used interchangeably, as are a and α

Thank you for noticing this inconsistency. We have now corrected the notation to explicitly define α and γ as the tuning parameters, which are selected as 0.25 and 2, respectively. The distinction between y and γ is also clarified.

**Reviewer #2:**

I found your study very interesting and relevant to the growing field of AI-driven forecasting for extreme ocean events.

We thank the reviewer for their kind comments and are glad that they deem this paper valuable to the field. We believe addressing their comments has made this paper more robust.

Below are some comments that I believe could strengthen your manuscript:

1.  Comparison with ocean models: While you acknowledge ocean models in the introduction, there is no direct comparison between your AI-based approach and a traditional numerical ocean model for MHW forecasting. A quantitative comparison—whether in terms of forecast accuracy, computational efficiency, or ability to capture physical processes—would provide valuable context and help clarify how AI complements or improves upon traditional approaches.

We appreciate this suggestion. In AI-based approaches, it is standard practice to compare the predictions of the AI model using a testing subset. This constitutes a quantitative evaluation of the AI-based solution relative to the model's performance in predicting MHWs without AI. For this reason, a portion of the dataset is excluded from training, ensuring the model has never encountered those

values. We have explicitly described this process in lines (209-214), and its results are reported in Section 3. Specifically, Fig. 4 and Tables 1–2 show the TPR, TNR, and FAR metrics, which are computed using the test dataset. These metrics compare the predictions from the ML model predictions to the "true" MHW labels, as these were derived from the RCSM model, by applying the MHW detection algorithm to the hindcast SST for all test days. The accuracy of these results is further discussed on lines 521-531 that read: "However, the assessment of the U-net CNN's forecast ability reveals distinct differences in predictive accuracy between the short-term (3-day) and longer-term (7-day) forecast horizon. While the 3-day forecast achieves the highest average TPR and TNR metrics overall, its discrepancy relative to the Persistence benchmark is modest, potentially due to the high autocorrelation of the SSTA in both the benchmark and the 3-day forecast (Fig. 6a). In contrast the 7-day forecast exhibits lower TNR and TPR values, with the performance gap between the benchmark and the U-net CNN being more pronounced (Fig. 6b). Thus, our results indicate an improved accuracy in shorter forecast horizons, in agreement with Bonino et al. (2023a). Despite demonstrating a decline in TPR from the 3-day to the 7-day forecast scenario, the strength of the U-net CNN lies in maintaining a higher TPR value than the Persistence benchmark in both forecast scenarios, suggesting an improved ability to predict MHWs across different temporal scales."

2. Resolution of the output: You mention that the final spatial resolution is downsampled to a 128x272 grid, but it would be helpful to explicitly state the corresponding resolution in kilometers. Additionally, how does the downsampling affect the model's ability to capture finer-scale MHW dynamics, particularly in regions with complex topography such as the Adriatic or Aegean Seas?

Thank you for your point. Due to computational constraints, we were unable to train the model at the original 6-8 km resolution. Thus, we performed a downsampling to an approximate resolution of 12-16 km, which is now explicitly mentioned in line 146. While downsampling itself may introduce uncertainties, these are better discussed in the added lines (538-547) that now read: "*Additionally, the downsampling applied to address the high computational demands of our method (see Sect. 2.2) resulted in a reduced spatial resolution of the data. This reduction likely affected the model's performance in regions with complex topography, such as the Aegean Sea and coastal areas. This is due to the reduction in spatial resolution (halved to 12–16 km) near the coast, which results in the averaging of values and the loss of high-resolution information in these regions. While we acknowledge this limitation, we prioritize reliable forecasting of MHWs across the entire Mediterranean Sea, understanding that higher-resolution models, though more accurate in these areas, also come with increased computational demands. This is also observed in Bonino et al. (2023a), where the authors also report a reduced forecast ability of their neural network model around the Adriatic Sea, Balearic islands, the Alboran and Aegean Sea*".

A key limitation of neural network methodologies is the inability to precisely identify the source of each inaccuracy. Fine-tuning these ML models relies on trial and error and comparisons with alternative approaches, as demonstrated in the Persistence benchmark section (section 2.5) and the sensitivity analysis (section 3.2) conducted in this study.

For example, uncertainties in our results could also arise from excluding wind, and perhaps other atmospheric variables, from our input dataset, from the imbalanced input dataset, as well as from accumulated model errors in regions with complex topography. We now discuss these, in added lines (545-557), that read:

*Given that ocean circulation along the coast is primarily driven by local winds and can be influenced by offshore currents near complex topographic features, the exclusion of winds as an input dataset may have*

*further reduced the forecast accuracy of our ML model in shallow coastal areas, where SST variations become more complicated (Berthou et al., 2024; Liu et al., 2025). However, Bonino et al. (2023a) found a weak dependence of the SST on wind speed across all the Mediterranean sub-basins they considered, by calculating the Mutual Information index prior to applying the ML method. While incorporating atmospheric variables into the training process could thus potentially enhance the model's ability to capture broader climatic influences on MHW occurrence, ultimately, we selected a limited set of training variables, in order to enhance the model's simplicity and replicability across both the training and prediction phases, following Sun et al. (2023).*

3. Uncertainty Quantification: Your study provides a robust evaluation of model performance, but there is limited discussion on uncertainty quantification. Given the stochastic nature of neural networks, have you assessed the sensitivity of your predictions to different training datasets, hyperparameter choices, or initial conditions? Methods such as ensemble modeling could provide insights into the confidence of the forecasts.

We appreciate the reviewer's comments regarding uncertainty quantification. As highlighted in Tables 1 and 2, our study performs a sensitivity analysis where models were retrained with different numbers of input variables. In most cases, the stability of the TPR, TNR, CM and FAR metrics remains high, except when an essential variable is entirely removed. We believe this provides a measure of robustness regarding input data choices. Additionally, in the supplementary material we show results for a different training-testing dataset splitting selection and we comment on the results on lines 559-573 that now read as:

*"It is important to note that the results of this study are based on the "straight split" methodology, where the early years of the dataset are used for training and the final years for testing. Given that ML models trained on recent data often achieve higher accuracy, due to the influence from recent climate patterns (recency effect; Lam et al., 2023), we have also explored the impact of reversing the training and testing dataset order. In particular, we carried out a sensitivity test, defined as the "opposite split", using the years 2013-2017 for training and 1982–1986 for testing, to assess whether our model's predictive skill is dataset-dependent or driven by climate change-induced temperature (see Supplementary Material Table S1). For the 7-day forecast horizon, we find improved TNR and TPR metrics when more recent data are inserted in the training dataset. This aligns with the findings of Lam et al. (2023), where recent climate trends, including the increased frequency of MHWs, were shown to enhance model effectiveness by mitigating issues of data scarcity and imbalance in the training datasets. However, for the 3-day forecast, we find a deterioration of both metrics, compared to the outputs of the "straight split" methodology (see Supplementary Material Table S1). This may be due to the longer duration of recent MHWs, which are less frequent in earlier years (Oliver et al., 2018) and thus may be poorly represented in the training dataset. In this study, we thus used the "straight split" methodology, as the training of a neural network with historical data to forecast information in the future reflects a more realistic approach. "*

Regarding different initial conditions, we do not have control over the circulation models which were used as inputs for our ML case study. These were kindly provided by the Med-CORDEX initiative, described in section 2.2, and we could not alter initial conditions to retrain with different models.

While ensemble modeling would, indeed, offer further insights into prediction confidence, computational constraints limited our ability to explore this in the current study. We now acknowledge and discuss this as a potential avenue for future research in lines 612-620, that read as:

*"As computational capabilities advance, ensemble models, such as those used in time series regression (Bertsimas and Boussioux, 2023) could also improve CNN-based forecasting of MHWs, especially for long-term model predictions that are typically hindered by error propagation. Future research should consider training ML models on observed or remotely sensed data, despite their limitations (Abdelmajeed and Juszczak, 2024), as well as hybrid approaches, that combine the physical consistency of traditional models with the speed and adaptability of ML methods (Bonino et al., 2023b). Given that the accuracy and reliability of many ML models is compromised by their violation of fundamental physical principles (Chen et al., 2023), future efforts should also focus on addressing this limitation, by incorporating physical laws (Desai and Strachan, 2021) or analytical equations (Zanetta et al., 2023), approaches that, though in early stages, show promise."*

4. Generalization Beyond the Mediterranean: You mention that the proposed method could be applied to other regions, but it would be helpful to discuss potential challenges in doing so. For example, would a model trained on Mediterranean SST anomalies generalize well to other basins with different oceanographic characteristics (e.g., stronger currents, different stratification, or more extreme variability)? A brief discussion of how the model could be adapted or retrained for different environments would be valuable.

We thank the reviewer for raising this point. The current trained CNN model is specific to the Mediterranean region due to the spatiotemporal characteristics of the dataset used for training. However, the methodology itself is adaptable and could be retrained using region-specific datasets. Given the Mediterranean Sea is a region where diverse circulation and thermohaline patterns can be observed, we can infer that the methodology can be effectively applied elsewhere. For instance, a similar AI technique has already been employed in the South China Sea (Sun et al., 2023), though without incorporating the novel loss function and attention mechanisms introduced in our work. We have added and amended a part of the discussion on this aspect to acknowledge the challenges and possibilities of generalizing our approach to other regions. (lines 586-593) that now read as:

*"Given the model's success in predicting MHWs using a minimum input of variables in a region with diverse thermohaline and circulation patterns, such as the Mediterranean Sea (Benincasa et al., 2024), it is reasonable to assume that our methodology can be generalized to other basins and case studies. While a similar ML approach has demonstrated comparable forecasting performance in areas with similar data availability (Sun et al., 2023), factors such as grid size and computational resources may also influence the training process of the ML model. Notably, the primary challenge in applying the U-Net CNN for predicting MHWs in this study was achieving a balance between predictive accuracy and computational efficiency, an important consideration for the application of all ML methods. "*

Overall, this study presents a promising application of deep learning for MHW forecasting, and I appreciate the detailed methodology and validation process. Addressing these points could further enhance the robustness and impact of your work.

We thank the reviewer for their positive feedback. We believe that addressing their feedback has significantly strengthened the robustness of the paper.

**Editor comments:**

Line 53, abbreviation "DL" should not be used, because it is used only 3 time in the manuscript.

Thank you for this suggestion. We have now changed it to "Deep Learning" throughout.

Line 164, it seems that local winds have not been considered as an input variable. This may partially account for the relatively poor ML model performance in the coastal ocean areas, such as the Aegean Sea and the Adriatic Sea. It is mentioned later in the paper that air temperature has been tested as an input variable, but not much improvement in MHW forecast. Have you tested winds as an input?

Thank you for raising this point. Following a correlation analysis of SST and mixed layer heat budget terms (not shown), we adopted an approach similar to Sun et al. (2023) selecting a minimal set of SST-correlated variables to maintain strong predictive performance and enhance replicability. Wind data was not included in the training phase. We now acknowledge its potential value for future work in lines (547-556). However, we would like to note here, that Bonino et al. (2023) found SST to have a week dependence on wind speed in a similar study for the Mediterranean Sea, demonstrating that the amount of information that can be gained from one variable by observing the other is low. We now discuss these, in added lines (545-557), that read:

*"This is also observed in Bonino et al. (2023a), where the authors also report a reduced forecast ability of their neural network model around the Adriatic Sea, Balearic islands, the Alboran and Aegean Sea. Given that ocean circulation along the coast is primarily driven by local winds and can be influenced by offshore currents near complex topographic features, the exclusion of winds as an input dataset may have further reduced the forecast accuracy of our ML model in shallow coastal areas, where SST variations become more complicated (Berthou et al., 2024; Liu et al., 2025). However, Bonino et al. (2023a) found a weak dependence of the SST on wind speed across all the Mediterranean sub-basins they considered, by calculating the Mutual Information index prior to applying the ML method. While incorporating atmospheric variables into the training process could thus potentially enhance the model's ability to capture broader climatic influences on MHW occurrence, ultimately, we selected a limited set of training variables, in order to enhance the model's simplicity and replicability across both the training and prediction phases, following Sun et al. (2023)."*

Line 220, should "NN" be 'CNN' o r a new acronym?

It was implied to be the neural network from the CNN acronym, because the overfitting that is commented is a challenge in general in neural networks. It is amended for readers' clarity and spelled out entirely.

Line 559 – 569, another possible reason could be the excluding of winds as an input the ML model. This is because coastal ocean circulation is mostly driven by local winds. Also, ocean circulation may be influenced by offshore currents forcing at the locations of complicated topographic features. These may not be properly represented in the ML model. Thus, temperature changes (and MHW changes) become more complicated in shallow coastal oceans (e.g., Berthou et al., 2024; Liu et al., 2025). It would be good to include these in the discussion.

References:

Berthou, S., Renshaw, R., Smyth, T., Tinker, J., Grist, J. P., Wihsgott, J. U., et al.: Exceptional atmospheric conditions in June 2023 generated a northwest European marine heatwave which contributed to breaking land temperature records, *Communications Earth & Environment*, 5(1), 287. https://doi.org/10.1038/s43247-024-01413-8, 2024.

Liu, Y., Weisberg, R.H., Sorinas, L., Law, J.A., Nickerson, A.K.: Rapid intensification of Hurricane Ian in relation to anomalously warm subsurface water on the wide continental shelf, *Geophysical Research Letters*, 52, e2024GL113192, https://doi.org/10.1029/2024GL113192, 2025.

We appreciate the Editor's suggestion on the potential impact of excluding wind as an input variable. Indeed, coastal ocean circulation is predominantly driven by local winds and offshore currents may influence circulation at locations with complex topographic features. We now acknowledge that these dynamics may not be fully captured by our model, potentially contributing to discrepancies in coastal regions such as the Aegean and Adriatic Seas. While our study followed an approach similar to Sun et al. (2023), where the number of input variables was kept minimal for model efficiency and replicability, we acknowledge that incorporating wind data could be investigated as part of a future research work. Although the study of Bonino et al. (2023) has shown that SST exhibits a low dependence on wind speed. Added discussion can be found in lines (545-557), that read:

*"This is also observed in Bonino et al. (2023a), where the authors also report a reduced forecast ability of their neural network model around the Adriatic Sea, Balearic islands, the Alboran and Aegean Sea. Given that ocean circulation along the coast is primarily driven by local winds and can be influenced by offshore currents near complex topographic features, the exclusion of winds as an input dataset may have further reduced the forecast accuracy of our ML model in shallow coastal areas, where SST variations become more complicated (Berthou et al., 2024; Liu et al., 2025). However, Bonino et al. (2023a) found a low correlation between the wind speed and SST in all the Mediterranean sub-basins they considered. To do that, prior to the ML method, the authors calculated the Mutual Information index using SST and several variables of interest, indicating that the amount of information that can be gained from one variable by observing the other is low. While incorporating atmospheric variables into the training process could thus potentially enhance the model's ability to capture broader climatic influences on MHW occurrence, ultimately, we selected a limited set of training variables, in order to enhance the model's simplicity and replicability across both the training and prediction phases, following Sun et al. (2023). "*

References:

Bonino, G., Galimberti, G., Masina, S., McAdam, R., and Clementi, E.: Machine learning methods to predict Sea Surface Temperature and Marine Heatwave occurrence: a case study of the Mediterranean Sea, EGUsphere, 2023, 1-22, 10.5194/egusphere-2023-1847, 2023.

Sun, W., Zhou, S., Yang, J., Gao, X., Ji, J., and Dong, C.: Artificial Intelligence Forecasting of Marine Heatwaves in the South China Sea Using a Combined U-Net and ConvLSTM System, 15, 4068, 2023.